# Multiscale Impact of Environmental and Socio-Economic Factors on Low Physical Fitness among Chinese Adolescents and Regionalized Coping Strategies

**DOI:** 10.3390/ijerph192013504

**Published:** 2022-10-19

**Authors:** Zihan Tong, Zhenxing Kong, Xiao Jia, Hanyue Zhang, Yimin Zhang

**Affiliations:** 1Key Laboratory of Exercise and Physical Fitness, Ministry of Education, Beijing Sport University, Beijing 100084, China; 2Institute of Physical Education, Northeast Normal University, Changchun 130024, China

**Keywords:** physical fitness, public health, environmental factors, socio-economic factors, spatial heterogeneity, multi-scale geographic weighted regression (MGWR), coping strategies, China

## Abstract

As low physical fitness in adolescents increases their risk of all-cause mortality in future adulthood as well as regional public health budgets, many scholars have studied the factors influencing physical fitness in adolescents. However, the spatial non-stationarity and scale between physical fitness and influencing factors in adolescents are often neglected. To rectify this situation, this study constructed a multi-scale geographically weighted regression model based on data from the China National Student Fitness Survey and the China Statistical Yearbook in 2018 to investigate the spatial patterns of factors influencing low physical fitness among adolescents. The results showed that the influencing factors for measuring the physical fitness of Chinese adolescents had significant spatial heterogeneity and multi-scale effects. The local R^2^ values were relatively low in the western region of China. Consideration should be given to increasing the lifestyle and ethnic and cultural characteristics of local residents when selecting influencing factors in the future. The physical fitness of men was mainly influenced by socio-economic factors, while that of women was influenced by natural environmental factors. According to the different spatial distribution patterns of MGWR, this study suggests that each region should develop regionalized strategies to cope with the low physical fitness of adolescents, including taking advantage of the natural environment to develop physical fitness promotion projects, accelerating the upgrading of industrial structures in the north-eastern and western regions, and the need to remain cautious of rapid urbanization in the east.

## 1. Introduction

Physical fitness is considered to be a key component of adolescent development and overall health, and a predictor of future health [1,2,3]. Physical fitness has been divided into sport-specific and health-related physical fitness, which can be considered a multifaceted construct [4,5]. The latter primarily includes cardiorespiratory fitness (CRF) and musculoskeletal fitness (MSF) [6]. CRF reflects the body’s ability to transport and use oxygen for energy transfer to support muscle activity during physical activity. MSF reflects the ability of a muscle or group of muscles to exert maximum force, explosive contraction, or sustained fatigue-free contraction, and the ability to move joints through a full range of motion. Overall, health-related fitness facilitates the promotion and maintenance of health and well-being and reduces health risks due to low levels of physical activity [7]. In particular, physical fitness is also important because the physical fitness that a person builds during adolescence has a significant impact on their physical fitness in adulthood and throughout their formative years and will continue into their later lives [8,9]. In addition, physical fitness has a significant impact on physical development and the maintenance of normal weight [10]. CRF and MSF in adolescents are also significantly associated with overweight and obesity, mental health, cognitive level, academic performance, and bone health [11,12]. At the same time, in the context of health improvement, adolescents should focus on physical fitness [13]. Enhancing the physical fitness of children and adolescents is an important means of enhancing health across the lifespan [5]. Low physical fitness has also been found to be associated with many chronic diseases and medical conditions [14]. There is a significant association between both low CRF and MSF and increased risk of obesity, self-esteem deficits, cardiometabolic disease, and all-cause mortality in early adulthood [15,16]. Regular physical fitness testing programs are important for initiating responses to negative trends in youth fitness. It can also be used to evaluate the implementation of current public health policies, adapt decisions to changes in the health of the population, and forecast the future burden of disease in the region [17].

Yet, the current situation is cause for concern with an overall downward trend in adolescent physical fitness over the last few decades [18,19]. The aerobic capacity of adolescents worldwide has almost universally decreased, while the body mass index has increased [20,21]. Available data show that 74% of Czech children have “below normal” muscle fitness and less than half of US adolescents are considered to be at a healthy level of physical fitness [22,23]. Physical fitness development is not consistent across countries [14]. Even different regions of the same country show physical imbalances depending on the development of natural, economic, and social determinants of health in each region. Previous studies have developed test instruments to assess the physical fitness of specific populations within individual states in the USA [24,25]. However, the variability and inconsistency of the data available across states are considerable, and the results suggest that it is not possible to directly compare fitness adaptations across states [26]. China’s latest report also points out that although the physical fitness of Chinese adolescent students has shown a steady improvement over the three years 2016–2018, the development of physical fitness among adolescent students in different regions and grades in China is uneven and inadequate [27].

Since the unevenness of physical fitness across regions involves all aspects of social development and will likely affect the evaluation of local region health indices, it is crucial to study the impact mechanisms of regional differences in physical fitness and to develop effective intervention countermeasures against low physical fitness [28,29]. Previous studies have elucidated the effects of socioeconomic status, family kinship, lifestyle, natural environment, and geographic location on adolescent health from a variety of perspectives, but there is a relative lack of research that explicitly uses spatial statistical analysis to explore the mechanisms influencing adolescent physical fitness. As a result, there is still insufficient progress on whether or how spatial location influences the development of adolescent physical fitness, or on spatial location-specific interventions for specific provinces to promote adolescent physical fitness. China is located in the eastern part of Asia and is a vast country, with great variation in the natural and social environments of different regions. Therefore, considering the possible spatial heterogeneity of physical fitness influencing mechanisms, it is likely that there are different combinations of influencing factors on the physical fitness status of students in different geographical locations. Compared to global regression models based on ordinary least squares (OLS), geographically weighted regression (GWR) models are more suitable for revealing health impact mechanisms and have been widely used in the field of public health [30,31,32]. However, GWR models are still more limited in their scale of action, with the assumption that all spatial processes of local coefficients are at the same scale, which does not reflect well the spatial scale differences between multiple influences and can capture more noise, causing bias in the process of model construction [33]. As an extension of the GWR framework, the MGWR model uses a different spatial smoothing determination method for each independent variable, which in turn generates a different bandwidth of variables, improving the geographically weighted regression model so that the model constructed for the sample not only observes spatial heterogeneity, but also enables the model to identify changes at different spatial scales, and is therefore gradually being applied in empirical studies of various impact mechanisms [34,35,36].

In summary, to fill these gaps, this paper takes junior high school students from 31 provinces in China who participated in the 2018 China National Student Physical Fitness Survey as the research object. It also adopts the student physical fitness failure rate as an indicator to measure the low physical fitness status of adolescents in each region and constructs a multi-scale geographically weighted regression (MGWR) model including environmental factors and socio-economic factors. The results of the model are used as the basis for proposing regionalized interventions to address the low physical fitness of Chinese adolescents. This will provide a theoretical reference for the further improvement of regional scientific fitness guidance programs, the prevention and control of physical decline, and the construction of health-centered and policy-oriented urban and environmental planning. Specifically, we will address the following research questions.
Should spatial heterogeneity and spatial scale variation be considered between the current low physical fitness and the influencing factors in Chinese adolescents?How do we regionalize low physical fitness Coping Strategies based on these possible spatially variable relationships?

The description of this paper is structured as follows. In the second section, we provide a review of the literature on physical fitness as well as spatial statistical models. In the third section, we provide details on the data sources and the chosen methods. In the fourth section, we present the main results around the spatial scale and spatial patterns of the influencing factors. In the fifth section, we discuss the results obtained and give recommendations for appropriate regionalized Coping Strategies. In the sixth section, we summarize the whole study and elaborate on the prospects and limitations.

## 2. Literature Review

### 2.1. The Concept of Physical Fitness

The term physical fitness has been conceptualized slightly differently in different periods and countries due to changes in the context of the times and the social environment in which it is used. In the United States, it has been argued that physical fitness is a state that allows a person to be fully physically and mentally conditioned to participate in the act of physical activity, thereby effectively reducing health risks. Physical fitness includes both health status and quality of movement [37]. Some European countries have subdivided physical fitness into elements such as physical form, physical function, motor quality, and quality of work, defining it as the quality of the body [38]. In Japan, work on the physical fitness of children and young people has also reached a relatively mature stage, defining physical fitness as physical strength, which mainly includes defensive strength to be able to defend and mobility strength to be able to adapt to the environment [39]. In China, scholars in sports science and related fields such as public health have also been exploring the concept of physical fitness and the various elements it contains. In the early days of the People’s Republic of China, the government established the fundamental task of improving the health of the people by enhancing their physical fitness through sport. With the advancement of the times and the maturity of relevant technology, the theory of physical fitness has gradually formed a distinctive ideological system and theoretical foundation thanks to the efforts of scholars from various disciplines [40]. Physical fitness is defined in China as the quality of the human body, a comprehensive and relatively stable characteristic of the body’s morphological structure, physiological functions, and psychological factors, expressed on a genetic and acquired basis.

### 2.2. A Study of the Factors Influencing Physical Fitness

Although physical fitness is, to a certain extent, genetically determined, it may be influenced by a combination of factors as it has multiple structural components. Scholars have synthesized previous studies on the factors influencing health in an attempt to examine the mechanisms influencing physical fitness from different perspectives. Socio-economic factors play an important role in the development of adolescent fitness, and there is evidence that the socio-economic status of parents influences the physical fitness of their children [41]. In a study from Spain, children’s obesity and musculoskeletal fitness are positively correlated with their parents’ levels of education [42]. A survey of approximately 2600 German children and adolescents showed that children with migrant backgrounds and lower socio-economic status were less active, which led to lower levels of physical fitness [43]. In another cross-sectional study, in a group of approximately 1400 German children and adolescents aged 7 to 16 years, the authors found higher socio-economic status was associated with higher physical fitness status [44]. More specifically, children and adolescents with higher socio-economic status had significantly higher levels of physical fitness than the reference sample, whereas children and adolescents with lower socio-economic status did not have significantly different levels of physical fitness from the reference sample. More recently, in a longitudinal study, significant differences in physical fitness were found between children with and without an immigrant background, with children who had an immigrant background performing significantly lower physical fitness at all time points tested [45]. Environmental factors are another important risk factor for low physical fitness in youths. There is evidence that the detection of obesity is lower among children living permanently in urban areas than in rural areas [46,47]. However, in studies in the Greek region, the findings were reversed for levels of adolescent fitness in urban and rural areas [48,49]. In China, Gui et al. found that for adolescents in ethnic minority areas, the influencing factors included individual lifestyle behaviors and school, family, and community environments [50]; Zhang et al. grouped the main factors affecting the physical fitness development of Chinese primary and secondary school students into multiple categories, with school environment factors accounting for the majority [51].

### 2.3. Spatial Heterogeneity and Spatial Scale

Most current research assumes that the way in which various factors affect physical fitness or health is spatially constant, resulting in a global model for assessing physical fitness and health. Such OLS-based global regression models lack the necessary flexibility in their practical application to examine the spatial heterogeneity of factors influencing physical fitness [52,53]. By incorporating spatial locations or geographical coordinates into the model, local regression models can be generated to better explain the mechanisms of physical fitness influence. Recently, Wang et al. used a geographically weighted regression to construct a model of China’s national physical fitness with indicators such as the natural environment and economic and social development to identify key factors that may have an impact on physical fitness [54]. He et al. found a significant spatially variable relationship between academic achievement and physical fitness and socioeconomic factors in Texas district schools by constructing a GWR model of the three and suggested that policies be issued to address key factors in specific districts based on this spatially variable profile [55]. Dwivedi et al. used OLS and GWR models to analyze the effects of commercial building indicators on adult physical fitness and chronic non-communicable diseases in each US county and found significant geographical differences in these relationships [56]. Yet, while these studies provide a preliminary exploration of the spatially variable effects of physical fitness determinants using geographically weighted regression models, they do not focus on the spatial scale of such variable effects. Spatial scale is a fundamental area of research in interdisciplinary disciplines, such as geography and spatially integrated social sciences. Goodchild proposed in 2000 that scale is the most important topic in geographic information science [57]. McMaster and Sheppard argued that scale is the essence of all geographical retrievals [58]. On the one hand, different types of social processes often correspond to different spatial scales, and there is a significant difference in scale between global and local spatial processes [59]. On the other hand, a certain social phenomenon may often be determined by a combination of spatial processes at different scales. Fotheringham et al. proposed multi-scale geographically weighted regression (MGWR) [33]. This method regresses each independent variable using its own optimal bandwidth, thus basically solving the problem of different scales and bandwidths of different variables. The application of MGWR in the field of public health has also been increasing in recent years. Cupido et al. examined the spatial variation in the relationship between mortality and various correlates by constructing multi-scale geographically weighted regression models using mortality data for each county in the United States, and their results showed that geographically weighted regression models incorporating spatial scales have higher explanatory power and predictive ability [60]. Oshan et al. analyzed adult obesity rates in Arizona by constructing OLS, GWR, and MGWR models, respectively, and concluded that MGWR was richer in the quantitative expression of determinants of obesity rates than the other two types of models [61]. The application of MGWR to infectious disease prevention and response is also of interest. According to Mollalo et al. who used models such as MGWR to analyze the spread of COVID-19 across the US, MGWR has the lowest AICc and higher explanatory power than other models, and the introduction of spatial scales can assist administrative units at all levels in making different responses [62].

In summary, scholars have studied the relationship between physical fitness and health, influencing factors, and spatial statistical analysis methods from different perspectives, using different data sources and analysis methods. However, there is a lack of research on the mechanisms of influence and regionalized responses to the weak physical fitness status of adolescents at the national scale in China, as well as a lack of results discussing the spatial patterns of the influencing factors through multi-scale geographically weighted regression models in the intersection of adolescent health and the environment. To fill these gaps, we used the MGWR model to examine the spatially variable effects of factors influencing youth physical fitness across 31 provinces in mainland China and propose regionalized coping strategies, based on data from the 2018 China National Student Physical Fitness Survey.

## 3. Materials and Methods

### 3.1. Studying Setting and Participants

The study area is the 31 province-level units in China, and the data are presented in the format of 1:1,000,000 electronic maps, provided by the China National Public Service Platform for Geographic Information, including surface elements of provinces, prefectures, and counties, as well as line elements of the nine-dashed line and the islands in the South China Sea, in readable and correctable vector files. China is located in the eastern part of Asia and Europe, on the west coast of the Pacific Ocean, and is a coastal country. The total land area is approximately 9.6 million square kilometers. The topography is high in the west and low in the east, with a stepped distribution. China spans a huge area in the south-east and north-west, has a varied climatic environment, possesses almost all types of terrain, and has developed diverse human and economic and social phenomena under its complex and diverse land cover information (Figure 1). China’s total population was 141.26 million at the end of the year 2021, forming a spatial distribution pattern of population in the provinces with Han Chinese as the main group and ethnic minorities developing together. As the world’s second-largest economy, China’s National Bureau of Statistics preliminarily calculated that China’s gross domestic product reached RMB 270,178 billion in the first quarter of 2022, an increase of 4.8% year-on-year. China not only has huge geographical differences in its natural geographical landscape, but also in its human and economic development, with significant geographical differentiation and agglomeration patterns [63], and it is of great theoretical and practical importance as a study area to understand the physical development of youth (data from https://www.webmap.cn/main.do?method=index/ (accessed on 4 August 2022)).

Subjects for this paper were selected from junior secondary school students from 31 provinces who participated in the 2018 China National Student Physical Fitness Survey. As a major project in China’s national school education effort, the results of the annual physical fitness survey are submitted in the form of a report to the Ministry of Education and feedback to the provincial education departments [64,65]. The Chinese mainstream media publicizes the annual student physical fitness survey upon its completion, and this work has made an outstanding contribution to completely reversing the decline in the physical fitness of China’s adolescents [66,67]. The detailed design of China’s National Student Physical Fitness and Health Survey project can be found in detail in the previous study [68,69]. In this study, students from 31 provinces across China were sampled as an overall sample through a stratified random whole-group sampling method. Based on the distribution of schools sampled in each locality, the sample was divided into four categories of students by grade level: Urban (male and female) and rural (male and female) for equal sampling. Each province was divided into 4 categories of sampling areas, and one prefecture-level city was selected from each category using a random method, and one junior high school built in an urban area and one junior high school built in a rural area were selected in that prefecture. In total, eight schools in four urban areas and four rural areas were sampled in each province. The number of students sampled was two randomly selected classes per grade level and a valid sample size of at least 60 students per grade level was ensured. To facilitate data comparison, the number of students sampled in all schools was approximately equal, resulting in a sample size of approximately 1440 students per province, after which the samples were tested for physical fitness, and the raw data, corresponding scores, and overall physical fitness scores were obtained for each sample. After data cleaning, a total of 48,922 junior high school students with complete information on gender, age, location, and physical fitness-related indicators were included in this study.

### 3.2. Data Declaration and Variable Selection

#### 3.2.1. The Dependent Variable

The dependent variable of this study was the failure rate of regional junior high school students’ physical fitness test in 31 provinces of China in 2018 based on the National student physical fitness standard. The National Student Physical Fitness Standard is an evaluation standard for the physical health testing status and exercise effects of Chinese adolescent students. The concept of health includes physical fitness, mental health, and social adjustment. The National Standard for Student Physical Fitness covers the category of student physical fitness, which is closely related to school sports. In order to define its connotation and avoid confusion with the three-dimensional concept of health, the term “physical fitness” is used as the definitive term for “health” to indicate its connotation. In mainland China, physical fitness tests mainly include body form, body function, and exercise quality. The body form is measured by height and weight to obtain the Body Mass Index (BMI), which reflects the growth and nutritional status of secondary school students. Body function is measured by spirometry, which reflects the working capacity of the respiratory system of the body. Exercise quality generally refers to the strength, speed, endurance, agility, flexibility and other physical abilities of the human body, which are reflected in the 1-min rope skipping, 50 m*8 round-trip running, 50 m running, sitting forward bend, standing long jump, pull-ups, 1-min sit-ups, 1000 m running (male), and 800 m running (female) tests, respectively. The testing staff received pre-service training and assessment tests, and the quality of on-site testing met the requirements. The statistics of the failure rate of the physical fitness test are evaluated based on the total score of the National Student Physical Fitness Standard, with ≤59.9 being a failure. The total score is composed of the sum of the standard score and an additional score, which is 120 points. The standard score is composed of the sum of the product of the scores and weights of each individual indicator, which is 100 points, among which the individual indicators and weights of the standard score for secondary school students include the body mass index (15%), lung capacity (15%), 50 m run (20%), sitting forward bend (10%), standing long jump (10%), pull-ups/1 min Sit-ups (10%), and 1000 m run for male/800 m run for female (20%). The additional scores are determined based on the actual test results and are worth 20 points. The additional scores for students are for pull-ups and the 1000 m run for the male group and 1-min sit-ups and the 800 m run for the female group, with a range of 10 points for each indicator.

#### 3.2.2. The Independent Variable

Regarding the choice of independent variables, this study attempts to explain the low physical fitness status of adolescents at the provincial scale in China in terms of different dimensions. Around the characteristics and influencing factors of physical fitness mentioned in the introduction, a series of indicators are listed that may have an impact on the situation of low physical fitness of Chinese youth (Table 1 and Table 2). As a composite concept, physical fitness has a wide range of influencing factors and needs to take into account regional indicators that are representative of each province in China, so the indicators for this study were selected from previous studies and existing established evaluation systems [63,70,71]. The indicators of regional characteristics such as the environment as well as socio-economic development selected for this study are based on attribute data from the China Statistical Yearbook 2018, China Environment Statistical Yearbook 2018, China Education Statistical Yearbook 2018, China Social Statistical Yearbook 2018, China’s 7th Population Census Bulletin, China’s provincial altitude barometric tables, and provincial statistical yearbooks for the year 2018.

### 3.3. Modelling Methods and Interpretation

#### 3.3.1. The Geographically Weighted Regression Model

Current research on the factors influencing adolescent physical fitness is mainly based on OLS; however, the global OLS assumes that the regression coefficients are the same for each sample point, ignoring the presence of spatial heterogeneity. Therefore, it does not clearly explain the spatial relationship between indicators of adolescent physical disadvantage and regional characteristics. As one of the main tools to deal with spatial heterogeneity, GWR was proposed by Fotheringham et al. [72,73]. The method is based on the idea of local regression analysis and variable parameters and is theoretically based on non-parametric methods of locally weighted regression such as curve fitting and smoothing, where the spatial location of the data is embedded in the regression parameters, and point-by-point parameter estimation is carried out using the locally weighted least squares methods, so as to study the regression relationships over space. Therefore, GWR has an advantage over OLS in dealing with the non-stationarity of spatial relationships [74].

The classical linear model, which is the basis of the geographically weighted regression model, is formulated as:(1)$$y=\overset{m}{\underset{j=0}{\sum }}\beta_{j}x_{j}+\varepsilon$$

The equation for the geographically weighted regression model is:(2)$$y_{i}=\overset{m}{\underset{j=0}{\sum }}\beta_{j}(u_{i},v_{i})x_{ij}+\varepsilon_{i}$$
where $\left(u_{i},v_{i}\right)$ are the coordinates of point. The estimates of the geographically weighted regression model can be given by the weighted least-squares method to give the results.
(3)$$\hat{\beta }(u_{i},v_{i})=\left(X^{\prime }W(u_{i},v_{i}\right)X{)}^{-1}X^{\prime }W(u_{i},v_{i})y$$
where X represents the matrix of independent variables, y represents the vector of dependent variables, and $W(u_{i},v_{i})$ represents the weight matrix, which is related to the spatial location. The main differences between geographically weighted regression models and classical global linear models are as follows: While traditional regression models have parameters that are constant in the global space, geographically weighted regression is a local model where the coefficients are obtained by regressing samples around the observation point, allowing the parameters to vary spatially. At the same time, geographically weighted regression can be used as a method for making statistical inferences about spatially varying relationships, thus shifting its focus to validation analysis. When the parameters are a function of the spatial location, this results in a model with insufficient degrees of freedom and more parameters than samples. According to the first law of geography, the closer the samples are to each other, the more similar the totality of the sample is, and thus the closer the estimate from the geographically weighted regression is to the true value, whereas points that are far away may be from a completely different totality. Therefore, to mitigate the effects of parameter drift, a spatial weighting matrix needs to be defined first to determine the appropriate neighborhood points around the regression point before carrying out geographically weighted regression. The geographically weighted regression defines a spatial weight matrix for each spatial unit i with the following equation.
(4)$$W_{i}=W_{\left(u_{i},v_{i}\right)}=\left\{\begin{array}{ccc} W_{i1} & 0 & 0\\ 0 & \ddots & 0\\ 0 & 0 & W_{in} \end{array}\right\}$$
where n represents the number in the study sample.

#### 3.3.2. The Multiscale Geographically Weighted Regression Model

The geographically weighted regression model specifies the same optimal bandwidth for each variable, which usually reflects the average of the optimal bandwidths for all independent variables. MGWR is one of the frontier methods for revealing the multi-scale dynamics and processes behind various human economic and social phenomena. Compared to GWR, MGWR has three main important improvements. First, allowing different levels of spatial smoothing for each covariate addresses the shortcomings of geographically weighted regression models [33,75]. Second, these covariate-specific bandwidths can be used as indicators of the spatial scale at which each spatial process acts. Third, the multi-bandwidth approach produces a closer real and useful model of the spatial process. Multi-scale geo-weighted regressions are more intuitive and easier to interpret than spatially variable coefficient models. Due to spatial heterogeneity, the influence of the explanatory variables of adolescent physical disadvantage in this study may vary across space and scales; therefore, this study uses MGWR to detect possible multi-scale influences in the low physical fitness of Chinese adolescents.

The multiscale geographically weighted regression model is as follows.
(5)$$y_{i}=\overset{m}{\underset{j=0}{\sum }}\beta_{bw_{j}}(u_{i},v_{i})x_{ij}+\varepsilon_{i}$$

In this equation, $y_{i}$ is the physical fitness of students in region j, $\beta_{bw_{j}}$ is the most appropriate sample bandwidth in region *j*, $x_{ij}$ is the observation of the j variable at i, and $\varepsilon_{i}$ is the random error term. The MGWR’s bandwidth selection criterion follows that of the classical GWR, using the corrected Akaike information criteria (AICc). AICc is a measure of model performance that helps to compare different regression models. Given the complexity of the model, a model with a lower AICc value will fit the observed data better and is useful for comparing models that apply to the same dependent variable and have different explanatory variables [76,77].
(6)$$AIC_{C}=2n\ln\left(\sigma \right)+n\ln\left(2\pi \right)+n\frac{n+\mathrm{tr}\left(S\right)}{n-2-\mathrm{tr}\left(S\right)}$$
where n represents the number of sample observations, σ represents the standard deviation of the error term, and tr(S), as a bandwidth function, is expressed as the trace value of the S matrix in the regression model.

The residual sum of squares (RSS) is used as one of the indicators to evaluate the effectiveness of the model fit. The smaller the observed value of the RSS, the better the model fits the observed data, and this value is also used in several other diagnostic measures, making it a very important parameter. In reliability testing of the MGWR model, researchers often use the R^2^, AICc, and RSS to perform a combined evaluation [78]. The convergence criterion for the proportional change in the residual sum of squares is used in this study and is given by the following equation.
(7)$$SOC_{RSS}=\left|\frac{RSS_{new}-RSS_{old}}{RSS_{new}}\right|$$

MGWR uses the parameters of GWR as initial estimates for iterative fit calibration, evaluating the best parameters as well as the bandwidth during each iteration until convergence ends when the difference in parameter estimates converges to a specified threshold, at which point the regression coefficients change by less than 10^−5^. $RSS_{new}$ and $RSS_{old}$ represent the sum of squared residuals from the current iteration and the previous iteration, respectively. The calculation of the MGWR model in this study was performed based on the MGWR2.2 software developed by the Spatial Analysis Research Center (SPARC) at the University of Arizona, Tempe, AZ, USA.

## 4. Results and Analysis

### 4.1. Models Accuracy and Scale Comparison

This section focuses on the effects of OLS, GWR, and MGWR, starting with a test of the problem of multicollinearity of the model. After pre-processing the relevant data information, we selected a variance inflation factor (VIF) < 10 as a condition to determine whether there is multicollinearity in the model [79]. Next, the normality of the influencing factors was tested to meet the basic requirements of the model [80]. It was found that some of the regional indicators met the normal distribution and could be directly incorporated into the regression model, while the distribution of some factors belonged to the long-tail distribution. We therefore take the natural logarithm for this part of the variable to ensure the data are more normally distributed [71]. Afterward, OLS, GWR, and MGWR models were constructed and evaluated for random distribution of residuals. The results of the three types of models are shown in Table 3 and Table 4. After considering the change in spatial local characteristics and the change in spatial scale, the R^2^ of the regression models improved significantly, and the R^2^ of the MGWR model was the highest among the three models. The AICc values of each model also showed a decreasing trend, with the MGWR model also having the lowest AICc value. This result implies that the MGWR model is a significant improvement over the OLS as well as the GWR.

In the MGWR model, as a type of local regression model, each spatially non-stationary sample point is computed to yield its own variable bandwidth with a local R^2^. The results for variable bandwidth (Table 3 and Table 4) reflect the degree of the differential effect of each variable in MGWR [81]. In the male group, the bandwidth of the GWR was 15.870, representing approximately 20.3% of the total number of observations. The spatial scales of the indicators varied in the MGWR for the male group: The INTERCEPT represents the status of the effect of different regional spatial locations on the failure rate of physical fitness, controlling for other independent variables [82]. The scale of effect is 7.890, which represents approximately 10.1% of the total number of observations and is much lower than the scale of effect of the other variables. In an average sense, this scale is close to the size of three provincial administrative units in China. Once the scale is exceeded, for example, when exploring the relationship between the physical fitness status of Chinese adolescents and their location, the coefficients will change dramatically, resulting in biased regression coefficients. The scale of action for RAIN is 25.140, accounting for 32.2% of the total number of observations, which is also relatively small in the regression model for the male group, indicating that the magnitude of the increase or decrease in the failure rate of physical fitness in the male group is influenced by annual precipitation in different provinces and municipalities, and varies widely in space. The URBAN in the male group has a bandwidth of 41.620, which is close to more than half of China’s land area and has a large scale of action, so the influence of this indicator on the physical fitness failure rate of Chinese junior high school males is stable in space and therefore less spatially heterogeneous. The effect scales for NF and ED were 78.090 and 78.070, respectively, which included all the number of observations and were global scales, and there was basically no spatial heterogeneity. In the results of the female group, the bandwidth of GWR is 78.100, and the model is judged as a global model, and it can be found that R^2^ and AICc do not change much against the OLS global model. The MGWR model for the female group shows that the action scale of ED is 24.420, accounting for approximately 31.2% of the total number of observation points, with a small scale and strong spatial heterogeneity. In addition, ELEVATION was also found to be spatially heterogeneous in the MGWR model for the female group, with variable bandwidth of 35.490, accounting for approximately 45.4% of the total number of observations, which is equivalent to half of China’s land area, and the coefficient is spatially stable. The other explanatory variables affecting the failure rate of physical fitness in the female group: The INTERCEPT, RAIN, NF, and BUILT are global scales, and there is basically no spatial heterogeneity.

The local R^2^ represents the strength of the actual interpretation of the selected indicators on the physical fitness of Chinese youth in each region. Figure 2 shows the spatial pattern of the results of the MGWR model by gender. The local R^2^ of the male group was distributed between 0.49 and 0.78, while the local R^2^ of the female group varied between 0.59 and 0.61, with a larger variation in the male group and a lower variation in the explanatory power of the indicator. The locational variation of the explanatory effect is large, while the female group is smooth with a low interval variation of explanatory power. In terms of the spatial pattern of the local R^2^, there was little difference between the gender groups, and the best explanatory power of the indicators for the physical failure rate of junior high school students was mainly concentrated in the three northeastern provinces, the Beijing-Tianjin-Hebei region, and the Yangtze River Delta region, which had larger R^2^ values and good fitting effects. However, in the northwest and southwest regions of China, the R^2^ values were relatively low, especially for the male group, where the R^2^ decreased by more than 0.2. This suggests that there may be an unaccounted-for omission in this region. At this point, the first research question posed in this study in the introduction to Part I has been addressed. That is, there is spatial heterogeneity between the low physical fitness of Chinese adolescents and the influencing factors and their relationship varies across spatial scales.

### 4.2. Spatial Pattern Analysis of Impact Factors

In order to reveal the spatial distribution of the coefficients of the impact factors in the MGWR model in different regions, we selected the statistically significant (*p* < 0.05) coefficients of the indicators in the model and classified them by gender and made thematic maps with the support of GIS software. In the MGWR results, the regression coefficients of RAIN and NF were significant in each gender group, while the regression coefficient of URBAN was significant only in the male group and ELEVATION was significant only in the female group, the results of which are shown in Table 5 and Table 6.

Firstly, RAIN (regional annual precipitation) was statistically significant in all gender groups in this study’s MGWR model, with a negative effect on the FR (Failure rate of provincial junior high school students in physical fitness tests). In the model for the male group, the coefficient of variation of the effect was close to −0.011, which indicates that for every 1 mm increase in annual precipitation in each region of China, the rate of youth physical fitness failure in that region decreased by close to 0.011%. The MGWR model for the female group shows that the influence of the annual precipitation coefficient varies around −0.006, which is slightly lower than for the male group, suggesting that the impact on the failure rate of female physical fitness in each region is an average increase of 0.006% for each 1 mm reduction in annual precipitation in that region. As shown in Figure 3a,b, the influence of precipitation on physical fitness in the MGWR model for both male and female groups showed a gradual increase from southeast to northwest, with the weaker changes in physical fitness concentrated in the areas southeast of the 800 mm isochronous rainfall line.

Secondly, the NF (non-farm structural ratio) has a negative effect on the FR1 (Failure rate of provincial junior high school male students in physical fitness tests), taking values between −1.015 and −1.014, with a mean of −1.015 and a standard deviation of 0.001, which indicates that for every 1 percentage point decrease in the non-farm structural ratio, the margin of increase in the failure rate of the physical fitness test of junior high school students in the region is 1.014–1.015 percentage points. In terms of the absolute value of the coefficient, the non-farm structural ratio has the largest value of all the influencing factors. The spatial distribution of the coefficients in Figure 3c shows that the influence of the non-farm structural ratio on the FR2 (Failure rate of provincial junior high school female students in physical fitness tests) generally tends to decrease slightly from west to east, and there is a clear stratification structure, with the extreme points of influence being Shanghai, Fujian, and Zhejiang provinces in the eastern coastal region, and the three northeastern provinces. From the MGWR model for the female group, the impact of the non-farm structural ratio on the low physical fitness of female adolescents is still negative, with a value of −0.629. The absolute value of the coefficient shows that the non-farm structural ratio has a relatively high intensity of influence, and because the bandwidth of this influence is on a global scale, the coefficient is relatively smooth and stable as it changes with spatial location, with little variation between regions. For every 1% increase in the non-farm structural ratio, the female youth physical fitness failure rate in the region decreases by 0.629%. Figure 3d reflects the spatial pattern of this coefficient across China’s significant provinces, showing an overall small increase in the negative influence of the non-farm structural ratio in the direction of the eastern to western regions. It is worth noting that, relative to the model for the male group, there is a tendency for the higher negative influence to spread towards the central regions in the model for the female group, suggesting that the influence of the non-farm structural ratio on middle-school females in all regions of China is gradually expanding. The highest negative impact of the non-farm structure ratio is in the western region of China.

Thirdly, URBAN (regional urbanization rates) are statistically significant social development influencers in the MGWR model of male adolescents with low physical fitness in China. Figure 3e shows the spatial distribution of the influence of urbanization rate, which shows a gradual increase from western to eastern China, and there is more variability between neighboring bordering regions. The regions most affected by urbanization are the three northeastern provinces and the coastal regions of Shanghai, Zhejiang, and Fujian. The effect of urbanization on the low physical fitness of male adolescents in each region was positive, and the risk of male adolescent students having low physical fitness increased with the increase in urbanization rate. The coefficients ranged from 0.237 to 0.241, with a mean value of 0.240, meaning that for every 1% increase in the urbanization rate, the regional increase in the risk of male adolescent physical fitness ranged from 0.237 to 0.241 percentage points, with the average increase varying by region to 0.24%.

Finally, ELEVATION (provincial average elevation) was a significant natural environmental influence on the MGWR model of physical fitness failure rate for the female group. For the MGWR model of the physical fitness failure rate of Chinese junior high school females, the effect of elevation ranged from −5.751 to −5.630, with a mean value of −5.672, which had a negative effect on the physical fitness failure rate, and the absolute value of the coefficient was larger. This indicates that the physical fitness failure rate of junior high school females is deeply influenced by the average elevation, decreasing by an average of 5.67% with a 1m increase in elevation. As shown in Figure 3f, the negative influence of average elevation decreases from northwest to southeast of China.

Overall, the results of this section have clearly explained the spatial pattern of variables on the low physical fitness of Chinese adolescents. In the next section of this study, we will address the second research question raised in this paper, which is how to regionalize low physical fitness coping strategies based on these possible spatially variable relationships.

## 5. Discussion

This study found that the multi-scale geographically weighted regression (MGWR) was more reliable than ordinary least squares (OLS) and geographically weighted regression (GWR) for analyzing the factors influencing the low physical fitness of Chinese adolescents, and that the MGWR not only takes into account geospatial differences, but also calculates the weights of different variables through a kernel function, which determines the spatial scale of the influencing factors. Spatial scaling effectively avoids capturing too much noise and bias in inaccurate results. Therefore, whether the spatial scale of the influencing factors is considered will have a significant impact on the results and analysis of the model. The distribution of the spatial pattern of the local R^2^ shows that the introduction of spatial scaling significantly improves the model fit in some areas of China. For the western regions, especially the north-western and south-western parts of China, where the fit is weak, there are several explanations for this phenomenon: From a physical geographic point of view, the north-western and south-western regions are mostly located in highland terrain with altitudes of over 1000 m or in arid and semi-arid regions with an annual rainfall of less than 500 mm. In the south-east, most of the regions are in plains, hills, or low mountainous areas, and have abundant precipitation throughout the year. The complexity of topography and climate causes the north-west and south-west regions to be affected by a combination of various geographical environments [83]. From the perspective of social development, the coastal areas of the south-east are well supplied with water, have excellent conditions for industrial and urban development, have a high density of transport networks, have more convenient access to the outside world, and are highly urbanized, with a stable and robust overall development structure. Northwest and Southwest regions have relatively weak population carrying capacity and non-agricultural industries [84,85]. The distribution of Chinese minority populations in the northwest and southwest is complex, with differences in lifestyles and regional cultures [86,87]. In summary, the specific situation of students’ physical fitness in the northwest and southwest regions needs to be studied in greater depth, considering the regional complexity behind the environment and socio-economic, as well as the unevenness of the indicators of physical fitness due to different geographical locations, and the addition and deletion of influencing factors according to the actual situation.

### 5.1. Suggestions for Regionalized Coping Strategies

#### 5.1.1. Confronting the Differences in Adolescent Physical Fitness between Regions in China and Intervening Based on Geographic Conditions

Studies on health and climate and the natural environment have shown that health and the natural environment are inextricably linked [88,89]. As society develops, the natural environment is constantly changing the way humans produce and live, balancing the material and energy exchange between humans and the environment [28]. Precipitation is a measure of climate change, and this study found that annual regional precipitation has a statistically significant effect on human physical fitness among natural environmental factors. The results of the spatial pattern of precipitation influence, combined with the regional distribution of annual precipitation in China, show that the southeastern coastal areas are the first to receive water vapor from the southeast monsoon and are already rich in precipitation. These areas are less sensitive to precipitation as an indicator, so the marginal effect is diminished. In the northwest, due to its deep inland location, distance from the sea, and low rainfall, a moderate amount of precipitation can cause strong climatic changes in the region, resulting in a change in the coefficient of influence. Thus, the spatial pattern of coefficient influence shows a development that is opposite to the pattern of precipitation in China. In addition, this study found that the average elevation of the region was a statistically significant factor in the influence of natural environmental factors in the female group, and that an increase in altitude was associated with an improvement in the physical fitness of regional females. By combining previous studies, it was found that there is a highly significant correlation between human aerobic capacity and altitude, and that appropriate altitude can effectively enhance human aerobic capacity [90]. Children living at higher altitudes have higher lung function relative to children living at lower altitudes [91,92,93]. Studies have shown that after a period of acclimatization to the plateau, students’ CRF improved significantly, and adolescents living at relatively high altitudes had better motor coordination, which also corroborates our findings that the appropriate altitude is beneficial to adolescents’ physical performance [94,95]. The spatial pattern of the influence of altitude may be explained by the fact that although youth physical fitness failure rates are influenced by a combination of factors, they are currently largely dependent on socio-economic development conditions. For the north-western region of China, due to the relatively weak level of economic and social development, the low physical fitness of female adolescents in the provinces is mainly influenced by the natural environment. In the eastern coastal regions, the level of urbanization is high and economic conditions are relatively good, so the influence of altitude on the failure rate of physical fitness is somewhat reduced. When comparing the gender differences in physical fitness failure rates, for female adolescents, the natural environment in which they live is more closely related to their physical health. In conclusion, it is important to address the differences in the physical fitness of young people in different regions of China due to the natural environment, and to pay more attention to the role and influence of the natural environment in promoting the physical fitness of young people. The program should be tailored to the needs of each region, such as the development of endurance sports in the plateau region, and the popularization of ice and snow sports in the northeast region.

#### 5.1.2. Accelerating the Upgrading of Industrial Structures in the Northeast and West of China to Build a Health-Friendly Environment

The non-farm industrial structure ratio, also known as the non-farm output ratio, is calculated as the ratio of the value of production in the secondary and tertiary sectors to the gross domestic product of the region and is the main regional development characteristic variable to measure the development of the industrial structure of a region [96]. As one of the key factors in the evaluation of social development and economic growth, industrial structure has always been highly valued by the Chinese government. According to the Petty–Clark Theorem, in the relationship between the evolution of industrial structure and economic development, the increase in the ratio of the non-farm industrial structure is an important role as the industrial structure is continuously upgraded, therefore the proportion of non-farm industrial output to regional GDP is the top priority for the upgrade of industrial structure [97]. At present, China’s economy is growing at a medium-to-high speed, and the construction of “Healthy China” is accelerating. As an important part of the national strategy of “Healthy China”, the development of students’ physical fitness, along with the process of China’s economic development, is a topic of concern. The development of economic factors can have an impact on people’s physical fitness to some extent, and some studies have shown that a higher economy can lead to a lower prevalence of obesity, which corroborates the results we obtained [98,99,100]. From the perspective of this study, the non-farm industrial structure ratio has a positive effect on promoting the physical fitness of Chinese adolescents, while the distribution of the spatial pattern of the influence of the non-farm industrial establishment ratio across gender groups shows that, firstly, the industrial structure in the western region needs to be upgraded, and the local resource as well as environmental conditions, regional location, and accessibility should be considered. Secondly, the development experience of oasis cities in western China should be taken into account, and industries should be gradually concentrated in the oasis region, which is conducive to improving technological progress and upgrading industrial structure in the western region, narrowing the difference in economic development with that of southeastern China, and promoting the physical fitness of youth [83,101]. Due to the high proportion of agricultural industries in the three northeastern provinces of China caused by special historical reasons, the future improvement of students’ physical fitness in this region can start with the idea of accelerating industrial restructuring in order to stimulate regional economic and social development and cultivate a healthy environment adapted to the development of students’ physical fitness. The eastern region of China should maintain an open economy and make full use of the resource conditions of the domestic and international markets to continue to upgrade and transform the economic structure and drive the healthy development of student fitness in the surrounding areas. The current industrial structure and urbanization in China should shift the focus of development and construction from large cities on the eastern coast to inland cities and suburbs around cities, remote areas, and rural areas, and at the same time, strengthen the fitness environment and promote positive interaction between social institutions, families, school communities, and the interpersonal environment of young people.

#### 5.1.3. Undertaking City Building Focused on Fitness and Maintaining a Cautious Approach to the Rapid Urbanization of Eastern China

The urbanization rate is a measure of regional urbanization, and it is a major factor in the field of social development. According to the latest statistics from the National Development and Reform Commission, the urbanization rate of China’s resident population reached 64.72% by the end of 2021. However, population urbanization is only one measure of the degree of urbanization; the quality of urbanization development still needs to be measured in conjunction with the level of public services such as healthcare and social security. With the rapid increase in urbanization, it is worth noting the impact of urbanization on the physical fitness of young people. In this study, we found that the risk of failing physical fitness among male junior high school students was positively correlated with urbanization, and that it was more profoundly affected in the developed eastern regions. The results of this study are consistent with the findings of several scholars, such as the finding that urbanization leads to a decrease in physical activity in the population, which results in increased health risks. The frequency of MVPA (moderate and vigorous physical activity) was significantly lower among adolescents living in urban areas than rural adolescents [102], while lower physical activity led to a decrease in physical fitness [103,104]. Higher physical fitness is more prevalent in rural areas [105]. In addition, studies have found that the growth and development of children and adolescents have increased with urbanization, and that the detection of overweight and obesity among children and adolescents has also increased, resulting in a decline in their physical fitness [106]. However, some scholars have also shown that urbanization has a positive effect on the health of the population. Adolescents in urban areas had better results than rural adolescents in terms of muscle strength, speed agility, CRF, and explosive power [107]. Urbanization affects the health of the population by increasing the economic level of individuals and the social security system, and the infrastructure and social conditions of developed areas make it easier for residents to access fitness environments and sports facilities [108,109]. We speculate that this may be due to the fact that some of the regions are in the early stages of urbanization, which has led to a significant increase in the level of economic development of the regions, bringing more benefits and better living resources to the residents, as well as a more robust health care system, which has helped to improve the health of the residents and children and adolescents. However, the increasing level of urbanization also has a direct impact on people’s lifestyles and diets. As adolescents are at a sensitive stage of growth and development, they are also one of the key groups affected by urbanization and information technology. Changes in dietary structure and habits, as well as the excessive involvement of electronic culture in their lives, lead to problems such as sedentary myopia, which have a significant impact on the physical fitness of adolescents [110]. Due to the contradictory results of the currently available studies and the fact that relevant indicators for evaluating urbanization were not fully included in this investigation, no clear conclusions can be drawn on how urbanization affects health. As other studies have shown, the impact of urbanization on health may be influenced by human development indices [111]. For example, there may be cultural differences between studies, differences in the definition of urban and rural cultural environments, and differences in the assessment of physical fitness in each country. What is certain, however, is that rapid urbanization has some side effects, such as a reduction in environmental quality and air pollution [112]. At present, China is in the process of changing from high-speed urbanization to high-quality urbanization. It is necessary to carry out more in-depth and prudent research and planning measures on the problems that urbanization may bring, to promote a new urbanization strategy with people at its core, to fully reflect the idea of putting people first, and to scientifically promote the construction of healthy cities.

## 6. Conclusions

This study analyzed the influencing factors and spatial patterns of adolescent low physical fitness in China through the recently published MGWR model using natural and socioeconomic data aggregated from the 2018 China National Student Physical Fitness Survey and various Chinese statistics. The need to intervene in the spatial scales of influencing factors in conducting analyses of adolescent physical fitness issues is highlighted, and corresponding regionalized interventions are proposed. The results show that:Natural environmental indicators such as elevation and precipitation, as well as socio-economic indicators such as non-farm industrial structure ratio and urbanization rate, have a significant effect on low physical fitness among Chinese youth, and demonstrated that the effect is spatially heterogeneous and multi-scale. The application of the MGWR model may yield more reliable results compared to OLS and GWR in conducting future studies on the spatial influence mechanism of adolescent physical fitness status.The spatial pattern of the influence of each indicator on low physical fitness was revealed. For the male group, the regression coefficient for the urbanization rate was positive, and the non-farm structure ratio and precipitation reduced the incidence of low physical fitness among male adolescents. The spatial heterogeneity of annual precipitation is high. There is a degree of spatial heterogeneity in terms of the effect of urbanization rates. The impact of the non-farm structure ratio exists on a global scale. For female adolescents, the three main influencing factors are mean altitude, annual precipitation, and the non-farm structure ratio, which are statistically significant. Mean altitude has a moderate spatial scale of effect, and there is spatial heterogeneity in its influence. All three indicators had a negative effect on the model. The coefficient of influence of the natural environment is the largest in absolute value, indicating that, for women, the natural environment in which they live is more inextricably linked to their own physical fitness.The current uneven and inadequate development of the physical fitness status of youth in different regions of China still exists, and there is a need to strengthen the regional youth physical fitness as well as reduce the number of physically disadvantaged individuals. Different types of regional physical fitness status aggregation and different approaches should be adopted to achieve refined and differentiated management. The regional industrial structure and current urbanization should shift the focus of construction from large eastern coastal cities to inland cities and suburbs around cities, remote areas in the west, and the countryside in general. The carrying capacity of the resources and environment should be improved to fully embody the new concept of high-quality urbanization that is people-oriented and integrated with the revitalization of the countryside. It is necessary to face the fact that there are large regional differences in the physical fitness status of Chinese youth and address the factors that influence it and develop targeted scientific physical fitness guidance programs according to local conditions. A sub-regional urban planning mechanism should be established to strengthen inter-regional linkages and take full account of the imbalances in various indicators of physical fitness due to different geographical locations.

### Prospects and Limitations

The relationship between adolescent health and geographic space has always been an important topic that cannot be ignored by society and academia. The scope of this thesis covers all provinces in mainland China, and the sources of physical fitness data, various natural environments, and socio-economic influencing factors are of good scientific validity and representativeness. Based on this, this study constructed a multi-scale geographically weighted regression model to explore the multi-scale spatial influence mechanism of youth physical fitness status, which is a useful attempt in the context of current interdisciplinary development and promotes the future development of existing youth physical fitness models and spatial statistical models. In addition, this study proposes regionalized interventions based on the current state of the physical disadvantage of young people in different regions of China. Regionalized interventions can provide a basis for the future improvement of the physical fitness status of young people in China and the implementation of public health guidance program, as well as a reference for planning departments related to healthy cities and the environment, so this study also has a degree of practical significance. We believe that, in the future, the results of multi-scale geographically weighted regression models should be used to screen the key factors affecting youth physical fitness in each region and to conduct thematic regional classification and clustering studies on the theme of youth physical fitness. Furthermore, based on the differences in influencing factors between western and eastern China found in this study and the relatively low explanatory power of the model in northwest and southwest China, we also see a need to examine the association between physical fitness and regional folk culture among adolescents in China’s ethnic minority concentrations in the future. Finally, we argue that with the global outbreak of COVID-19, it is necessary to conduct a longitudinal spatio-temporal trend analysis using multi-year data related to adolescent physical fitness to quantify the direct and indirect impact of the epidemic on adolescent health in each region by comparing the changes in adolescent physical fitness before and after the epidemic at different spatial locations.

Despite the important methodological and theoretical and practical implications of this study, the following limitations remain unavoidable. Firstly, as China has been affected by COVID-19 in 2020, large-scale data collection on youth physical fitness surveying has taken a hit. Considering that there may be some degree of impact on adolescent fitness before and after the epidemic, cross-sectional data on adolescent fitness in 2018 were selected for this study and failed to demonstrate trends in the relationships between variables over time. Secondly, as the coverage scale of the current implementation program of the National Student Physical Fitness Survey is the provincial administrative units of China, the spatial unit included in this study is the corresponding 31 provinces of mainland China (excluding Hong Kong, Macao, and Taiwan). This may lead to changes in the configuration and number of spatial units in future studies of youth physical fitness at small local scales, such as prefecture-level cities or county-level cities, and the results of the model may change accordingly and overwrite some information. However, the choice of scale for processing geographically aggregated data is an issue that needs to be considered over time and is not specific to this study. Finally, as physical fitness is a composite construct, and as the geographical location and developmental status of different countries contribute to the complexity of the factors influencing physical fitness, the inclusion of the influencing factors and the results of the model in this study require further in-depth testing and research.

## Figures and Tables

**Figure 1 ijerph-19-13504-f001:**
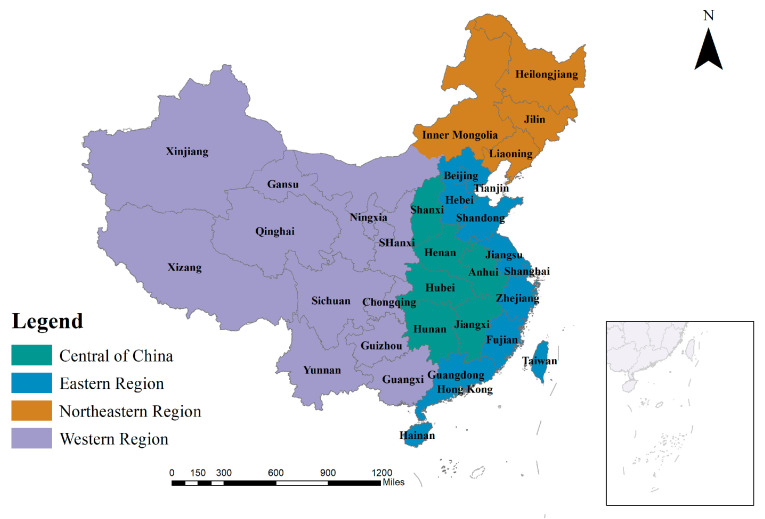
The study area: China’s 34 provincial administrative units and four economic regions.

**Figure 2 ijerph-19-13504-f002:**
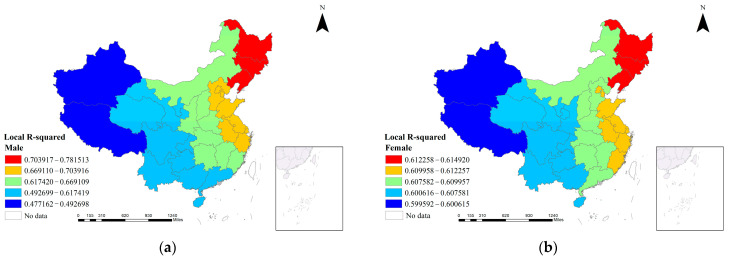
Spatial patterns of local R-squared in the MGWR. (**a**) Local R-squared spatial patterns of the male group in China, (**b**) local R-squared spatial patterns of the female group in China.

**Figure 3 ijerph-19-13504-f003:**
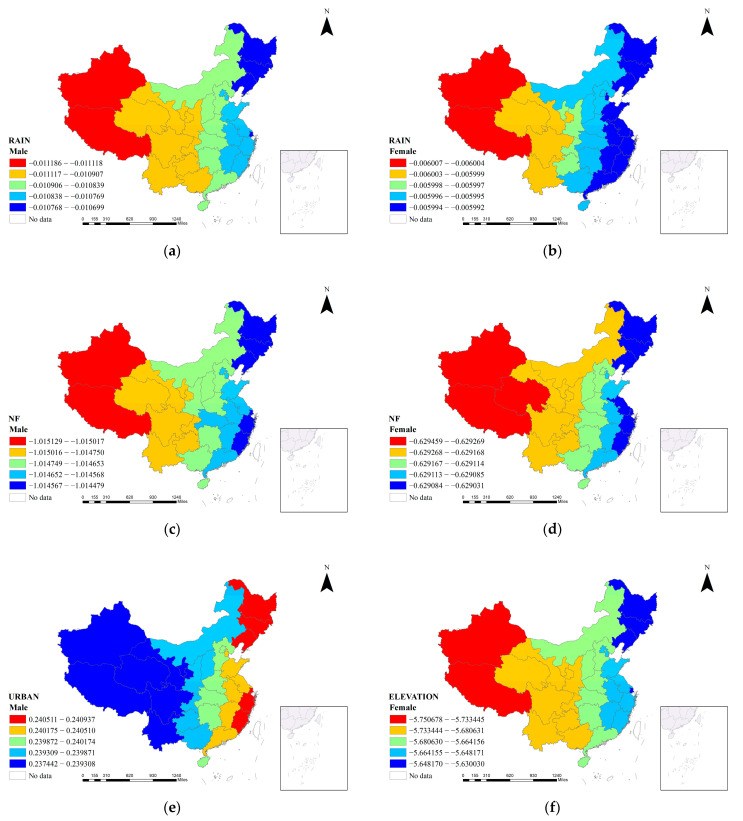
Spatial patterns of coefficients in the MGWR. (**a**) Annual precipitation in the male group; (**b**) annual precipitation in the female group; (**c**) non-farm output ratio in the male group; (**d**) nonfarm output ratio in the female group; (**e**) urbanization rate in the male group; (**f**) elevation in the female group.

**Table 1 ijerph-19-13504-t001:** A full list of dependent and independent variables.

	Variable Name	Description of Variable
Low physical fitness	FR1	Failure rate of provincial junior high school male students in physical fitness tests (%)
	FR2	Failure rate of provincial junior high school female students in physical fitness tests (%)
Environmental factors	HUMIDITY	Average of annual relative humidity in the province (%)
	ELEVATION	Provincial average elevation (meters)
	SUNLIGHT	Annual sunlight hours in the province (hours)
	RAIN	Total average of monthly precipitation during the year (mm)
	TEMP	Average of provincial temperatures during the year (°C)
	BUILT	Area of the provincial built-up region (Km^2^)
	NOISE	Average intensity of provincial ambient noise (dB)
	EMISSIONS	Annual emissions of provincial exhaust gases (ton)
Economic factors	GDP	The ratio of provincial GDP to provincial resident population
	ED	Economic density of the provincial districts (persons/km^2^)
	NF	The ratio of provincial non-farm output to GDP (%)
	INCOME	The ratio of provincial income to provincial resident population
Social factors	PD	Population density of the provincial districts (persons/km^2^)
	URBAN	The ratio of urban population to total population (%)
	EDUCATION	Average years of schooling in the province (years)
	ATHLETES	Number of athletes at provincial level 2 or above

**Table 2 ijerph-19-13504-t002:** Description and descriptive statistics of all the variables.

Variable Name	Mean	Std.Dev.	Min	Max
FR1	15.821	8.602	4.299	41.717
FR2	7.419	5.903	0.839	23.507
HUMIDITY	65.226	12.085	38.0	82.0
ELEVATION	536.581	859.563	3.0	3958.0
SUNLIGHT	2066.087	489.482	1059.7	3054.5
RAIN	957.590	509.031	280.2	2135.3
TEMP	14.5	5.02	5.1	24.4
BUILT	1885.677	1331.594	164	6036
NOISE	54.297	1.566	49.1	56.9
EMISSIONS	947,358.032	557,452.141	73,135	2,036,544
GDP	64,687.739	30,268.711	30,797	153,095
ED	0.455	1.042	0.001	5.716
NF	91.183	4.973	76.638	99.709
INCOME	28,166.106	11,279.025	17,286.06	64,183
PD	461.349	706.551	2.882	3928.571
URBAN	60.305	12.057	30.225	93.781
EDUCATION	9.879	0.973	6.75	12.64
ATHLETES	1498.903	935.362	54	4368

**Table 3 ijerph-19-13504-t003:** Comparison of fit results of low physical fitness regression models for male groups.

Indicators	OLS	GWR	MGWR
Residual sum of squares	978.796	752.620	697.158
R^2^	0.573	0.672	0.696
AICc	210.497	208.574	207.377
Bandwidths	—	15.870	7.890 (INTERCEPT)
			25.140 (RAIN)
			78.090 (NF)
			41.620 (URBAN)
			78.070 (ED)

**Table 4 ijerph-19-13504-t004:** Comparison of fit results of low physical fitness regression models for female groups.

Indicators	OLS	GWR	MGWR
Residual sum of squares	441.063	437.135	422.090
R^2^	0.592	0.595	0.609
AICc	189.155	189.175	188.657
Bandwidths	—	78.100	78.100 (INTERCEPT)
			35.490 (ELEVATION ^1^)
			78.100 (RAIN)
			24.420 (ED ^2^)
			78.100 (NF)
			78.100 (BUILT ^3^)

^1, 2, 3^ The above variables have been logarithmically calculated.

**Table 5 ijerph-19-13504-t005:** Statistical description of the MGWR model coefficient for male groups.

Variable Name	Mean	Std.Dev.	Min	Median	Max
INTERCEPT	104.694	2.095	95.208	105.245	107.159
RAIN	−0.011	0.000	−0.011	−0.011	−0.011
NF	−1.015	0.000	−1.015	−1.015	−1.014
URBAN	0.240	0.001	0.237	0.240	0.241
ED	−0.907	0.003	−0.915	−0.907	−0.903

**Table 6 ijerph-19-13504-t006:** Statistical description of the MGWR model coefficient for female groups.

Variable Name	Mean	Std.Dev.	Min	Median	Max
INTERCEPT	80.805	0.010	80.775	80.807	80.818
ELEVATION ^1^	−5.672	0.026	−5.751	−5.668	−5.630
RAIN	−0.006	0.000	−0.006	−0.006	−0.006
ED ^2^	−3.035	0.132	−3.266	−3.051	−2.642
NF	−0.629	0.000	−0.629	−0.629	−0.629
BUILT ^3^	−4.817	0.008	−4.845	−4.817	−4.804

^1, 2, 3^ The above variables have been logarithmically calculated.

## Data Availability

Not applicable.

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
