# Peer review of "Multiscale Impact of Environmental and Socio-Economic Factors on Low Physical Fitness among Chinese Adolescents and Regionalized Coping Strategies"

_ijerph, 2022, doi:10.3390/ijerph192013504_

Round 1
Reviewer 1 Report
Dear authors,
I hope you are well, and I would like to say that I liked your paper very much. If possible I would like you to address the following suggestions:
- Include a literature review section.
- Include a few lines on practical implications of the work, as well as future lines of research.
- Likewise, the methodology used seems to me to be correct. Regarding the conclusions, it would be interesting to expand it with future lines of research and go deeper into the practical implications.
I wish you luck in your publication.
Best regards
Author Response
Thank you for all the things you have done for us, and we have uploaded the revisions made as an attachment.
We wish you all the best and look forward to hearing from you.

Reviewer 2 Report
Current submission constructed a multi-scale geographically weighted regression (MGWR) model based on data from the China National Student Fitness Survey and the China Statistical Yearbook in 2018 to investigate the spatial patterns of factors influencing low physical fitness among adolescents. Please conduct the concerns below.
1. In the introduction, application of MGWR in public health is limited without reason(s). Then, why MGRW in current study will provide a theoretical reference that needs to describe in detail.
2. MGWR is more reliable than the ordinary least squares (OLS) or the geographically weighted regression (GWR) that needs the evidence.
3. The annual physical fitness survey submitted to the Ministry of Education is the main reference that remained un clear in current analysis.
4. Statistical description of the MGWR model coefficient needs the reliability or suitable reference(s) to support.
5. The distribution of Chinese minority populations in the northwest and southwest is complex, with differences in lifestyles and regional cultures. How to get the MGWR model be reliable?
6. Conclusions need to revise to be clear in brief with novelty.
7. The data collection on youth physical fitness surveying has been hit in current analysis. How to compensate it in the report?
Author Response

(The authors gave the same response as above.)
